# Differences between the Fittings of Dental Prostheses Produced by CAD-CAM and Laser Sintering Processes

**DOI:** 10.3390/jfb14020067

**Published:** 2023-01-26

**Authors:** Mariano Herrero-Climent, Miquel Punset, Meritxell Molmeneu, Aritza Brizuela, Javier Gil

**Affiliations:** 1Porto Dental Institute, Av. de Montevideu 810, 4150-518 Porto, Portugal; 2Biomaterials, Biomechanics and Tissue Engineering Group (BBT), Department of Materials Science and Engineering, Universitat Politècnica de Catalunya (UPC), Av. Edurad Maristany 16, 08019 Barcelona, Spain; 3Barcelona Research Centre in Multiscale Science and Engineering, Technical University of Catalonia (UPC), Av. Eduard Maristany, 10-14, 08019 Barcelona, Spain; 4Facultad de Odontología, Universidad Europea Miguel de Cervantes, C/del Padre Julio Chevalier 2., 47012 Valladolid, Spain; 5Bioengineering Institute of Technology, Faculty of Medicine and Health Sciences, International University of Catalonia, Josep Trueta s/n, 08195 Barcelona, Spain

**Keywords:** dental prosthesis, marginal gaps, CAD-CAM, laser sintering, CrCo, palatal fit, gingival fit

## Abstract

Digital dentistry and new techniques for the dental protheses’ suprastructure fabrication have undergone a great evolution in recent years, revolutionizing the quality of dental prostheses. The aim of this work is to determine whether the best horizontal marginal fit is provided by the CAD-CAM technique or by laser sintering. These values have been compared with the traditional casting technique. A total of 30 CAD-CAM models, 30 laser sintering models, and 10 casting models (as control) were fabricated. The structures realized with chromium–cobalt (CrCo) have been made by six different companies, always with the same model. Scanning electron microscopy with a high-precision image analysis system was used, and 10,000 measurements were taken for each model on the gingival (external) and palatal (internal) side. Thus, a total of 1,400,000 images were measured. It was determined that the CAD-CAM technique is the one that allows the best adjustments in the manufacturing methods studied. The laser sintering technique presents less adjustment, showing the presence of porosities and volume contraction defects due to solidification processes and heterogeneities in the chemical composition (coring). The technique with the worst adjustments is the casting technique, containing numerous defects in the suprastructure. The statistical analysis of results reflected the presence of statistically significant gap differences between the three manufacturing methods analyzed (*p* < 0.05), with the samples manufactured by CAD-CAM and by traditional casting processes being the ones that showed lower and higher values, respectively. No statistically significant differences in fit were observed between the palatal and gingival fit values, regardless of the manufacturing method used. No statistically significant differences in adjustment between the different manufacturing centers were found, regardless of the process used.

## 1. Introduction

The five properties that make a dental restoration a success are biocompatibility, esthetics, static and cyclic mechanical strength, good corrosion behavior, and marginal adaptation. Lack of marginal adaptation will result in the accumulation of food debris, plaque, and an increased risk of caries. In addition, the lack of fit will cause micromovement that will affect the surrounding tissues [1,2,3,4,5].

Acceptable gap values are considered to be between 50 and 200 μm, but there is no consensus on this aspect [2,3,5]. Manufacturing methods are improving processes to increase accuracy. However, a decrease in accuracy could be due to faster manufacturing speeds, reduced economic costs, saving materials to reduce costs, and increasing the speed at which the clinician has the prosthesis [6,7]. In the past, metallic restorations have been elaborated by using metal casting by means of the lost-wax technique. The casting process produced metal restorations with low mechanical properties, the presence of porosity, shrinkage, and lack of chemical homogeneity (coring) that affected their mechanical and corrosion properties, and in addition, the casting process was a very laborious and manual process [8,9,10]. This fact led to new technologies to avoid these drawbacks which made more reproducible and defect-free restorations.

One of these methods is computer-aided design/computer-aided manufacturing (CAD-CAM), which is based on machining. A more recent method is the metal laser sintering technology. These two methods eliminate many of the shortcomings of the casting system and make it more reproducible without such a high dependence on the technician performing the casting [11,12,13,14,15]. This has been a major breakthrough in digital dentistry [16,17,18]. The use of digital processes achieves better accuracy following the workflow stages: scanning, design, and manufacturing.

The synergy between new manufacturing technologies and digital dentistry has brought a revolution in implant dentistry in which the quality of restorations is increasing every day. In addition, the emergence of new materials has given another boost to the long-term performance of dental restorations [19,20].

In recent years, new alternative materials to nickel alloys have appeared in the field of dental restoration, in order to limit and/or eliminate the allergenic effects of nickel, among which the following stand out: ceramic materials such as yttria-stabilized zirconia, thermosetting polymers such as PEEK, and metals that no longer contain nickel [21,22]. Materials with improved mechanical properties and excellent corrosion resistance have been developed and have surpassed precious metals [21]. One of these alloys is CrCo, which has been increasingly applied due to its excellent results. Cobalt-based alloys are known for their strength, hardness, and resistance to corrosion. Chromium provides hardness and resilience and increases corrosion resistance when its concentration is between 16 and 20 wt% due to the formation of a chromium oxide which provides a passive film in the entire suprastructure. For aesthetic purposes, dental alloys used in fixed prostheses may have a porcelain veneer fired onto the cast substructure [23,24,25].

This study attempts to provide as an original element the determination of horizontal gaps in superstructures fabricated by CAD-CAM and laser sintering with the same model of suprastructure. It is the first study with such a large number of measurements. In addition, we were able to determine whether the differences found depend on the company that performs them, using six of the most prestigious ones.

The hypothesis of this study would point to the existence of significant differences in fit between the three manufacturing methods compared, regardless of the manufacturing center or company. The null hypothesis was that no significant differences would be found either between manufacturing methods or between manufacturing companies in terms of marginal gap adjustment.

## 2. Materials and Methods

### 2.1. Elaboration of Models

The master model was obtained from a patient who needed to restore several edentulous areas. Fixed prostheses needed to be replaced due to their failure, and the teeth also needed retreatment. The patient refused implant treatment of the edentulous areas and it was proposed to rehabilitate these areas with fixed prostheses. After the endodontic retreatments were performed, the tooth preparations were corrected, and intraoral impressions were taken with a Virtual^®^ Light Body Regular Set and Virtual^®^ Putty Base Regular polyvinylsiloxane (Ivoclar Vivadent AG, FL-9494 Schaan, Liechtenstein) to reproduce teeth 14 and 16 with metal–ceramic crown preparations as well as the pontic area of the edentulous 15 (Figure 1). The edentulous area was restored with a metal–ceramic fixed prosthesis with pontic in zone 15.

The polyvinylsiloxane impression was digitized using an ISCAN L1 SERIES laboratory scanner (IMETRIC 3D SA, Geneva, Switzerland). The model was fabricated in an epoxy resin using a STRATASYS dental 3D printer (GMBH, Gladbecj, Germany) (Figure 2).

#### The Tooth Preparations had the Following Characteristics:

Deep circumferential chamfer of 1.2 mm at the level of the cervical termination line-die height of about 6 mm.

Obtaining a silicone key for duplication of the reference model.

An addition silicone model (Wirosil^®^ duplicating silicone, BEGO GmbH & Co, Bremen, Germany) was used for the duplication of the reference model according to the manufacturer’s instructions. Components 1 and 2 of the duplicating silicone were mixed in a 1:1 ratio until a homogeneous color was obtained. The mixture was poured into a system-specific duplicating cuvette and 40 min elapsed to ensure complete polymerization of the duplicating material. The reference model was carefully removed from the mold created with the duplicating silicone (Figure 3).

### 2.2. Master Model Obtention

Duplication procedures were performed in a room where humidity and temperature conditions were within the limits of temperature (23–25 °C) and of relative humidity (70–80%). All duplication procedures and master models’ obtention for each structure were performed by a single operator.

The duplicating silicone key and an epoxy resin (Diemet-E, Erkodent^®^ Erich Kopp GmbH, Pfalzgrafenweiler, Germany) were used to obtain master models, following the manufacturer’s instructions. The resin, hardener, and spoonful’s powder (filler) were mixed in the same container according to the manufacturer’s instructions. The components were mixed until a homogeneous consistency and the resin was poured into the duplicating silicone key. The working time was about 15 min, and the epoxy resin set at the end of 8 h in a temperature range of 23–25 °C. At the end of resin polymerization, the duplicate model obtained was removed and the presence of pores or imperfections were evaluated (Figure 4). In cases where pores or any type of imperfection existed, they were discarded so as not to alter the samples. The procedure was repeated until a total of 30 duplicate models of the master reference model were obtained.

#### Confection of the Cr-Co Structures

These frameworks were not cemented. All structures were made of Cr-Co alloy (Dentaurum Gmbh & Co., Ispringer, Germany) (Figure 5) with the same chemical composition in all cases, which was determined by energy-dispersive X-ray spectroscopy (EDS) microanalysis. CoCr alloys are the most used in the dental restorations. One of the most interesting properties of this alloy is its wide melting temperature range that avoids distortion of the structures. The chemical composition in weight percentage is shown in Table 1.

The size of the Co-Cr powder particles (Remanium Star CL, Dentaurum, Ispringen, Germany) was 10–40 μm in diameter with a coefficient of thermal expansion of 14.1 10–6 K^−1^ and with a melting temperature around 1495 °C.

A total of 70 components were analyzed, at a rate of 10 components (n = 10) for each manufacturing plant evaluated. A total of 10 casted components were tested as a control group, together with 30 laser sintering components and 30 CAD-CAM components. The components formed by CAD-CAM and LS were manufactured at a rate of 10 components for each of the 3 different manufacturing sites used for each technique. The companies were Archimedes (La Seu d’Urgell, Spain), Createch Medical (Mendaro, Guipuzcoa, Spain), Straumann (Manohay Dental, Alcobendas, Spain), Avila-Mañas Laser (Madrid, Spain), Phibo (Setmenat, Barcelona, Spain), and Oral Design Center Galicia (Lugo, Spain).

The structures used to carry out the evaluations had the following dimensions (Figure 5):-The thickness was 0.5 mm over the entire surface of the structure.-The pontic of the 15 position had a convex shape on the cervical surface with a distance of 1 mm with respect to the edentulous ridge recorded on the master model.-At the level of the connectors, a total area of 5 mm^2^ corresponding to an apico-coronal length of 2.5 mm and a vestibule-palatal thickness of 2 mm was used.-It was applied virtually or by means of a spacer varnish with a thickness of 50 μm, located up to 1 mm from the termination line of the dies.

For CAD-CAM, the master model was scanned with an IMETRIC optical scanner (Courgenay, Switzerland), and the STL file obtained was transferred to a CAD software (EXOCAD, Darmstadt, Germany) in which the structure was designed incorporating the characteristics described. The STL file of the structure was sent to the Hyperdent CAM (Munchen, Germany) software and the structure was milled on a SAUER 10 5-axis milling machine (Isny im Allgäu, Germany). Model fit, dimensions, and volume of the structure were verified (Figure 6).

For laser sintering, the master model was scanned with an IMETRIC optical scanner (Courgenay, Switzerland), and the STL file obtained was transferred to a CAD software (EXOCAD) in which the structure was designed incorporating the characteristics described. The STL file of the structure was sent for sintering, using a Cr-Co powder (Cr-Co-Mo-W-Si) in a Yterbio laser machine, which applied powder layers with a thickness of 0.02 mm. Once the structure was finished, it was cleaned using a jet of 250 μm alumina oxide particles (Figure 7).

### 2.3. Determination of Adjustments

Samples were observed with a scanning electron microscope (SEM) under 20 KV potential conditions. The JSM-6400 (JEOL, Tokyo, Japan) scanning electron Microscope was used. This was equipped with an EDS (energy-dispersive X-ray spectroscopy, OXFORD model Xmax20, Oxford, UK), which allowed for the identification of the chemical composition by means of the acquisition of the characteristic X-ray emission of each chemical element.

To fix the structures inside the SEM chamber, models were anchored with 50 N load in the center of the pontic area to ensure proper fixation. A load was applied to avoid micromotions that could affect the measurements. The 50 N load could not cause plastic deformation or damage to the model (Figure 8).

The samples were observed at 5000 magnifications and placed with automated coordinate axes and angles. This system allowed the different “horizontal gaps” analysis and the comparison of the different samples fit at previously established points. Six areas of the specimens were checked, three on the outside of the specimen to check the vestibular fit and three on the inside to evaluate the lingual fit. Samples at 5000 magnifications were measured at 10,000 sample points for each suprastructure on the electron microscope with the image analysis system attached.

Minitab 16 Statistical Software (State College, PA, USA) was used to evaluate any statistically significant differences among samples. A normality test was performed in order to evaluate the normality of data. If *p* < 0.05, data were not normal and non-parametric tests were used; if *p* > 0.05, data were normal and parametric tests (Student’s *t*-test to compare 2 samples and ANOVA to compare more than 2 samples) were carried out. For all the tests were performed, if *p*-value > 0.05, no statistical differences were found.

## 3. Results

The results of the gaps are based on more than 10,000 measurements for each of the samples studied, which gives us a very accurate idea of the distances between the substrate and the model. Therefore, 10,000 measurements × 2 positions (external and internal) × 10 samples for each system × 7 different processes (3 CAD-CAM, 3 laser sintering, and 1 casting method) are 1,400,000 measurements. Figure 9 shows the gaps of the samples obtained by CAD-CAM for each of the suppliers. An image of the gaps and the machining lines on the surfaces can be seen. Figure 10 shows the gaps of the samples obtained by laser sintering. In this case, it can be seen how the shapes of the samples are more rounded due to the melting process that the laser exerts on the cobalt chrome. Figure 11 shows the gaps in the samples that have been cast, where some pores and some shrinkage of the volume of the material can be seen.

Figure 9 shows the separation values for each of the CAD-CAM processes studied, and Figure 10 shows the values for laser sintering as well as the one we used as a control, which is the traditional casting process (Figure 11). The values obtained of marginal gaps can be observed in Figure 12. 

For the different companies that produced the superstructures, the results showed the best fits for CAD-CAM compared to laser sintering. The cast samples showed the worst results. Differences were statistically significant for the CAD-CAM process with respect to LS and casting, with a *p* < 0.05. The CAD-CAM and LS showed statistically significant differences compared with the casting process, with a *p* < 0.0.05. LS-2 and LS-3 model gaps had statistically significant differences compared with all CAD-CAMs and casting, with a *p* < 0.05. The differences in LS-1 compared with CAD-CAMs were statistically significant, with a *p* < 0.05.

No statistically significant differences were seen in any case between internal and external gaps, with a *p* < 0.05.

## 4. Discussion

The machined specimens show a good fit due to the high-precision machine tools that currently have five axes and allow to reproduce very accurately the complicated shapes with sharp curvatures of the prostheses. A sample of an area of a part with good reproducibility is shown in Figure 9. The laser sintering samples have, as can be seen in Figure 10, a less precise finish due to the sintering processes that cause local welding of the particles leading to localized fusions [26,27,28,29,30].

Figure 13 shows the quality of the superstructure obtained by CAD-CAM, where no defects of inclusions of inorganic material, porosity, or defects of lack of chemical homogenization are observed, since these processes are carried out at low temperatures. Machining flaws and burrs can be seen in the angled areas, since the machining machines, even five-axis ones, sometimes do not give a perfect finish [31]. For this reason, very acute angled areas should be avoided, and scratches should be avoided as they can lead to the onset of fatigue cracks if the prosthesis is mechanically loaded with a high mechanical stress [32,33,34,35,36].

The laser sintering process shows a different surface finish than CAD-CAM, since sintering requires temperatures close to the melting points of CrCo and there are local fusions that allow joining the spheres of the material. Figure 14a shows this surface, in which the rounded shapes typical of the melting or microfusion processes can be appreciated. The compaction pressure in laser sintering makes the cohesion good. As is well known, the mechanical properties of a sintered material increase with the compaction pressure and sintering temperature. In Figure 14b it can be seen at higher magnifications the presence of a sphere that has not been able to cohere sufficiently, as well as different shades on the surface [36,37,38].

In Figure 15 it can be seen in greater detail these chemical heterogeneities that are called coring. This fact is due to variations in the chemical composition, as can be seen in the X-ray energy-dispersive diffractograms in which some areas richer in cobalt and other areas richer in chromium can be observed. Porosity can be observed in the laser sintering surfaces with spherical shape. These porosities are due to the expulsion of gases contained in the metal and released to the exterior [39].

The surface of the cast superstructures can be seen in Figure 16, where the walls are rounded as is normal in cast structures. Moreover, in this case, it can be seen more intensely that the laser sintering surfaces undergo volume shrinkage processes due to the smaller volume of the solid than that of the liquid, and this causes defects on the surface of the structure. Figure 17 shows a shrinkage in which the interior shows alumina residues coming from the molds in which the material has been cast. Therefore, with this methodology we also have inclusions of refractory materials from the mold in the melting processes. Much more frequently, porosities such as the one shown in Figure 17 will also be observed. All these defects affect the quality of the superstructure and justify the worse marginal fit values illustrated in Figure 13.

The porosity obtained in the cast samples is superior to the laser sintering samples, as can be seen in the polished surface of the material obtained by casting a significant number of pores (Figure 18). However, the sizes are small, approximately between 5 and 10 μm, and in principle should not significantly affect the mechanical properties of the structure [40].

This melting and solidification process provides the possibility of porosity formation in the samples, as can be seen in Figure 11, and also the defect of shrinkage caused by volume contraction in the transition from liquid to solid. This phenomenon generally occurs at the crystal boundaries, producing a decrease in the mechanical properties of flexural strength and fatigue behavior. This defect can be seen in Figure 17; in which it can be differentiated from porosity by the non-spherical shape of the defect [40,41,42].

The casting method is the one that presents the greatest number of defects, both in porosity and in the rejects, since the fusion is total. In addition to these defects, the prosthesis may contain residues of the ceramic coatings of the casting molds [42]. In addition, the differences in melting points of the alloying elements cause chemical heterogeneities in the prosthesis material, where there are areas richer in chromium and others in cobalt, as can be seen in the microanalyses of the laser sintering suprastructures in Figure 15. This phenomenon is likely to cause electrochemical corrosion problems. To avoid this problem, it is necessary to perform a homogenization annealing treatment to the prosthesis at a temperature of 400 to 600 °C for 30 min to achieve homogenization in the prosthesis and avoid zones with different electrochemical potentials inside the prosthesis.

The limitations of this study have been tried to be reduced by having the same model scanned and making the superstructure in different companies to avoid the effect of manufacturing. However, further studies remain to be conducted, such as determining the vertical gap or seeing how the gaps may evolve with cyclic masticatory loads [43], as well as the influence of the material from which the superstructure is fabricated.

## 5. Conclusions

According to the results obtained in this study, the improvement in the finishing quality of CrCo dental prostheses with the use of CAD-CAM and laser sintering techniques is observed, which has been reflected in the reduction in the manufacturing adjustment. This conclusion would confirm the fulfilment of the starting hypothesis of this study, given the existence of statistically significant gap differences between the three processes evaluated. It was possible to observe the defects produced by the laser sintering, such as volume contractions, porosity, and chemical heterogeneity in the CrCo used for the fabrication of the suprastructures. The traditional casting method is the one that presents the worse adjustments, because the number of defects is higher due to the metal solidification process. The comparative study between the two alternative techniques to traditional casting (CAD-CAM and laser sintering) shows the existence of statistically significant differences in fit between them, with the CAD-CAM technique presenting the optimum values of fit. This conclusion validates the main hypothesis initially put forward in this study. With respect to the study of different manufacturers and adjustments carried out with both alternative techniques evaluated, the statistical analysis of results shows no statistically significant differences in adjustment between the different manufacturing centers, regardless of the technique used, except for one company with high-precision machines. Such a conclusion partially fulfils the null hypothesis initially stated in this study. Regarding the orientation of the fit measurement performed during the study, the analysis of the results did not reveal the existence of statistically significant gap differences between the fit values determined in the palatal and gingival areas.

## Figures and Tables

**Figure 1 jfb-14-00067-f001:**
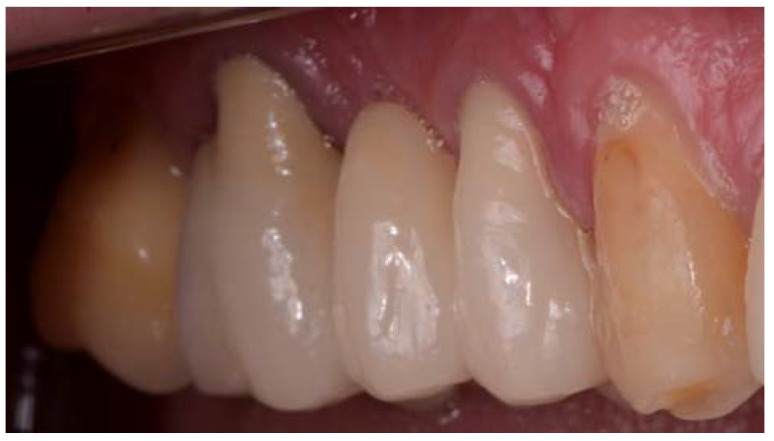
Fixed metal–ceramic prosthesis of teeth 15, 16, and 17 from which the master model was obtained.

**Figure 2 jfb-14-00067-f002:**
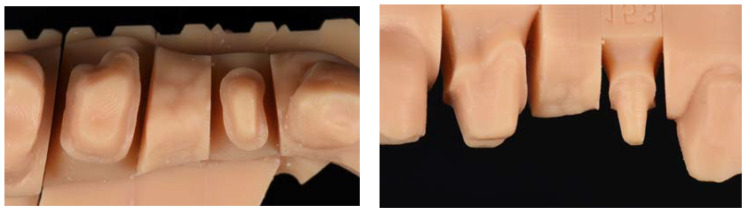
Master model from occlusal view showing the tooth preparations of 14 and 16 and the pontic area of 15.

**Figure 3 jfb-14-00067-f003:**
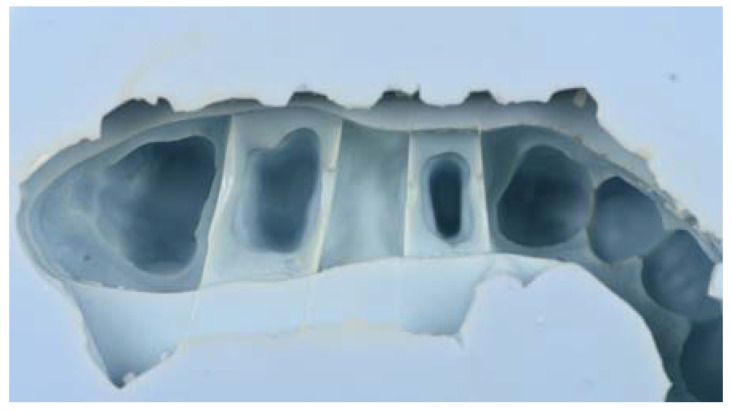
Photographic image of the silicone mold made from the master model.

**Figure 4 jfb-14-00067-f004:**
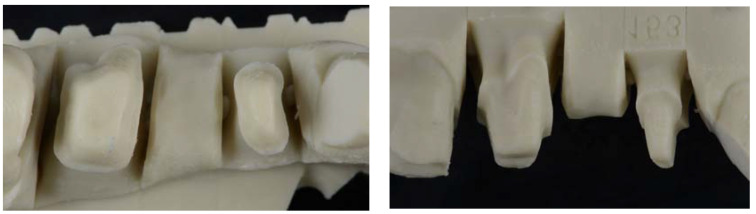
Images of duplicate model from occlusal and vestibular views showing the tooth preparations of 14 and 16, and the pontic area of 15.

**Figure 5 jfb-14-00067-f005:**
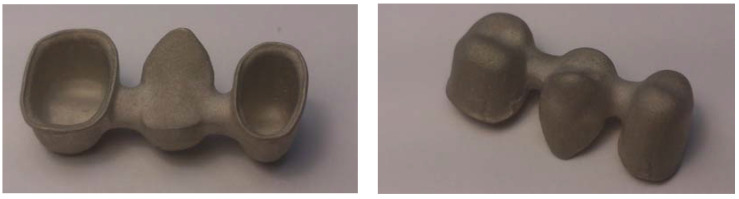
Images of the master model to be reproduced by all the manufacturing centers, with details of its top (**left**) and bottom (**right**) views.

**Figure 6 jfb-14-00067-f006:**
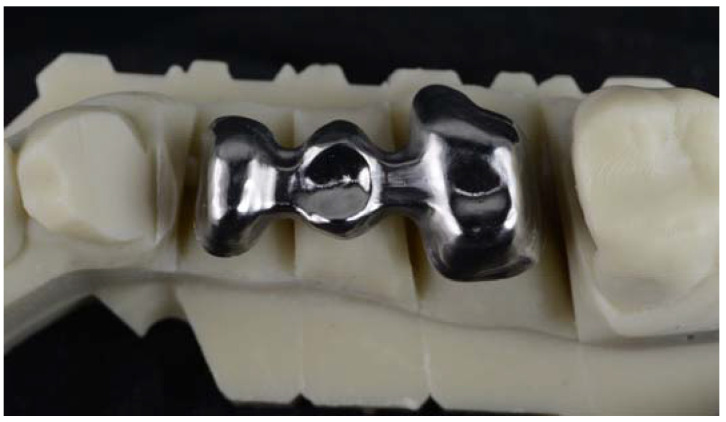
Image of a specimen manufactured by CAD-CAM (Archimedes Pro).

**Figure 7 jfb-14-00067-f007:**
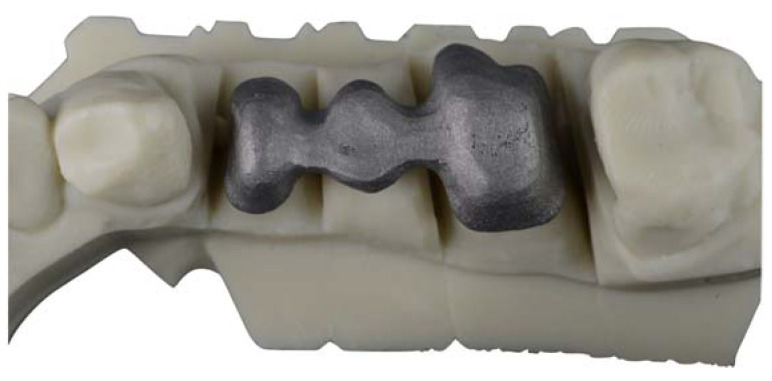
Image of a specimen manufactured by CAD-CAM laser sintering (Archimedes Pro).

**Figure 8 jfb-14-00067-f008:**
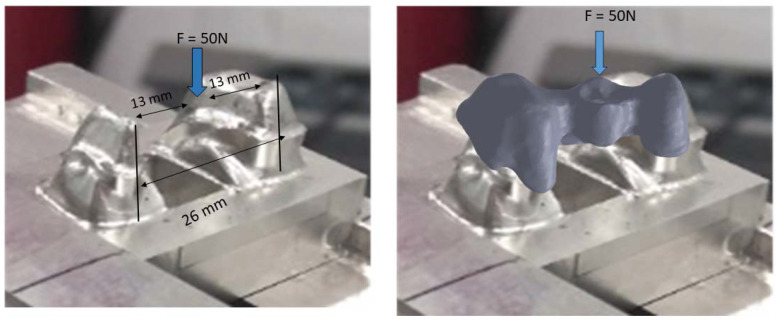
System for the fixation of the sample to measure the marginal gaps.

**Figure 9 jfb-14-00067-f009:**
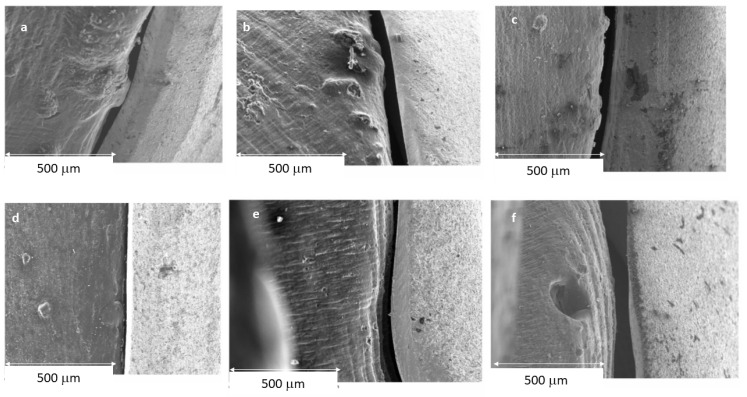
SEM micrographs of marginal gap determination in CAD-CAM manufactured specimens: (**a**) CAD-CAM-1 (internal); (**b**) CAD-CAM-2 (internal); (**c**) CAD-CAM-3 (internal); (**d**) CAD-CAM-1 (external); (**e**) CAD-CAM-2 (external); and (**f**) CAD-CAM-3 (external).

**Figure 10 jfb-14-00067-f010:**
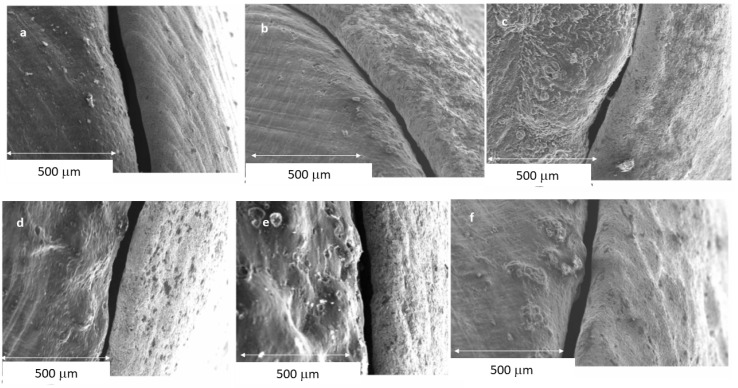
Representative SEM micrographs with detail of marginal gap determination in laser sintering specimens: (**a**) LS-1 (internal); (**b**) LS-2 (internal); (**c**) LS-3 (internal); (**d**) LS-1 (external); (**e**) LS-2 (external); and (**f**) LS-3 (external).

**Figure 11 jfb-14-00067-f011:**
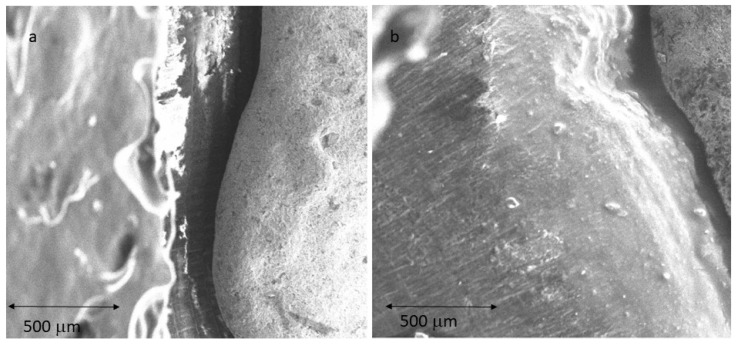
SEM micrographs with detail of marginal gap determination in casted specimens: (**a**) internal; (**b**) external.

**Figure 12 jfb-14-00067-f012:**
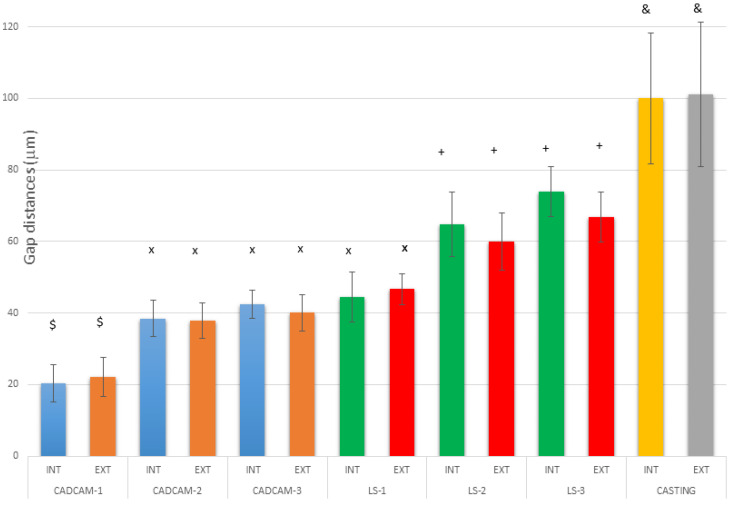
Results of marginal gaps—internal and external—for each system and for the different manufacturers. Each sign in the columns means that there are statistically significant differences between them with a *p*-value < 0.05.

**Figure 13 jfb-14-00067-f013:**
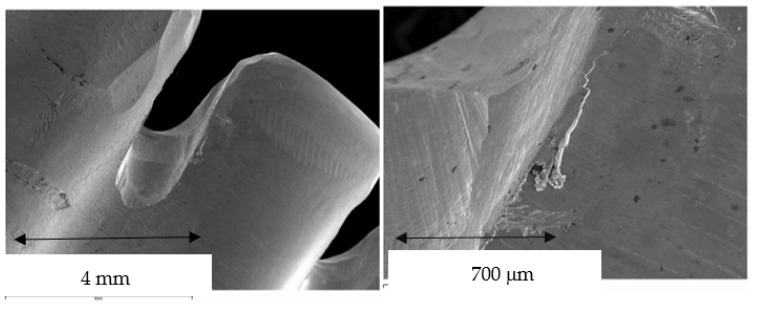
SEM micrographs of CAD-CAM machined specimens with details of small surface defects in the form of machining flaws (**left**) and burrs (**right**).

**Figure 14 jfb-14-00067-f014:**
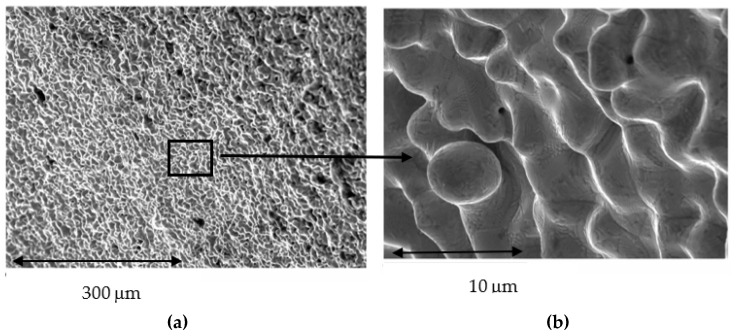
SEM micrograph of laser sintered specimens with details of small, partially melted powder particles adhered to typically rounded surface texture produced by laser microfusion processes (**a**); at higher magnifications (**b**).

**Figure 15 jfb-14-00067-f015:**
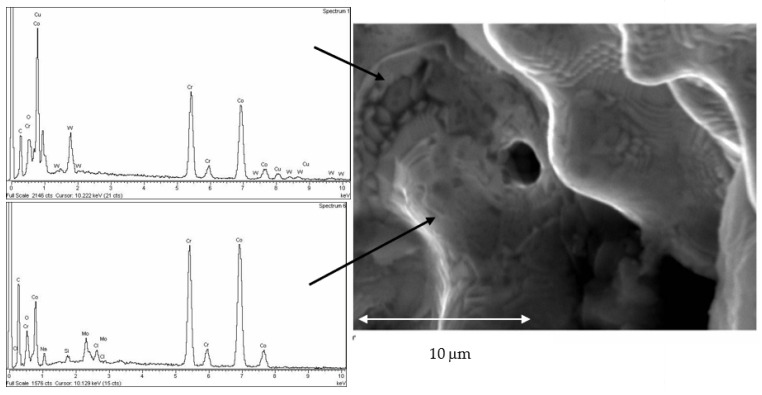
SEM micrograph of laser sintered surface with detail of both chemical segregations reflected as different chemical composition in different places (coring) and gas spherical microporosity.

**Figure 16 jfb-14-00067-f016:**
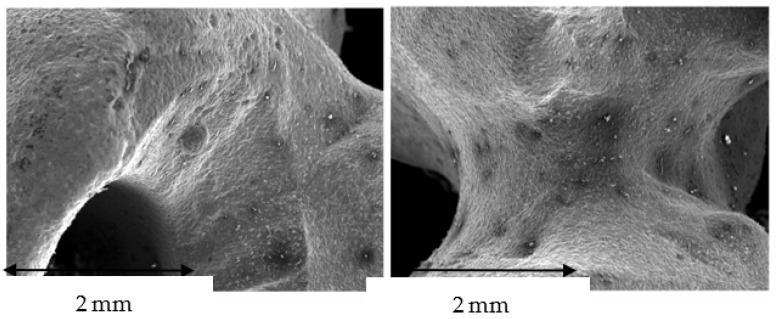
SEM micrographs of the surface of the suprastructure obtained by casting.

**Figure 17 jfb-14-00067-f017:**
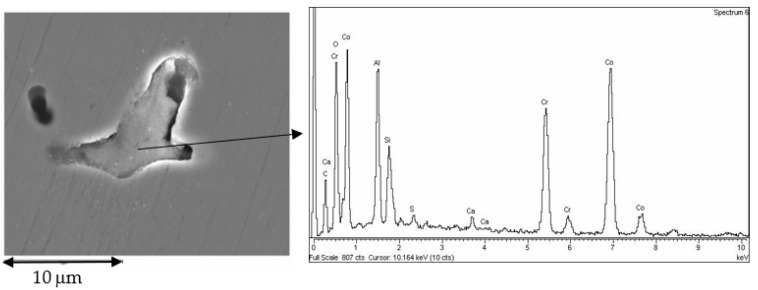
Defect due to volume contraction in the solidification process. The interior shows traces of alumina and silica refractory materials that have remained deposited. Energy-dispersive X-ray microanalysis showing the presence of Al and Si is illustrated.

**Figure 18 jfb-14-00067-f018:**
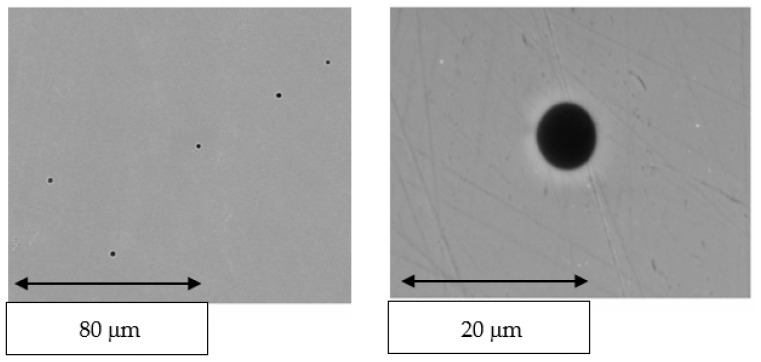
SEM micrographs showing porosity obtained in the casted suprastructures.

**Table 1 jfb-14-00067-t001:** Chemical composition of CoCr alloy (%wt): Co, cobaltium; Cr, chromium; W, wolframium; Si, silicon; C: carbonium; Nb: niobium.

Chemical Element	Co	Cr	W	Si	C	Nb
(%wt)	56.53 ± 2.11	27.11 ± 1.31	9.64 ± 0.79	1.27 ± 0.80	<1%	<1.5

## Data Availability

The data presented in this study are available on request from the corresponding author.

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
