# Peer review of "Differences between the Fittings of Dental Prostheses Produced by CAD-CAM and Laser Sintering Processes"

_jfb, 2023, doi:10.3390/jfb14020067_

Round 1

Reviewer 1 Report

Abstract:

·      Data including the p value should be added to the results section.

·      The Conclusion section should include more details regarding the results. It must be somewhat extended.

Introduction:

·      Reference should be added for "acceptable gap values" on lines 52-53.

·      PEEK materials appear to cause allergies in lines 75–77. It ought to be the opposite, though. The phrase needs to be rewritten. It is not understood.

·      The purpose sentence of the article on lines 87-90 should be reconstructed. It is not understood. The new sentence should have passive sentence structure, more formal expressions.

 Materials and methods: On line 207, Turkey analysis was written by mistake. It must be Tukey.

Results:

·      This section is very complex. It is not understood. It should be rewritten.

·      In particular, the titles describing figures 9,10,11 should be rewritten. Figures are incomprehensible. It should be stated in the figure titles that these images are SEM images.

·      The naming on the figures is confusing.

Discussion:

·      The titles of Figures 13-18 should be rewritten. Please write more clearly.

References: Reference 1, 6, 8, 10, 19, 35 and 40 are very old. Please replace what you can with new ones.

Author Response

REVIEWER 1

Dear Reviewer,

Thanks for taking the time to review our manuscript and suggest to us to improve our work by providing a lot more detail. We have done so, and we are now submitting a manuscript that not only addresses the points you specifically raised but also many others that we have considered in order to deliver what we think is a much-improved version of our work. This version includes more paragraphs, figure, English grammar revisions in all main sections, new references. Thanks a lot. We are looking forward to your comments.

Sincerely,

Francisco-Javier Gil Mur

Abstract:

  • Data including the p value should be added to the results section.

DONE: Conclusion section presented in abstract has been extended including more details regarding statistical results.

  • The Conclusion section should include more details regarding the results. It must be somewhat extended.

DONE: Conclusion section presented in abstract has been extended including more details regarding statistical results.

Introduction:

  • Reference should be added for "acceptable gap values" on lines 52-53.

Done.

  • PEEK materials appear to cause allergies in lines 75–77. It ought to be the opposite, though. The phrase needs to be rewritten. It is not understood.

DONE: This problematic phrase has been revised.

  • The purpose sentence of the article on lines 87-90 should be reconstructed. It is not understood. The new sentence should have passive sentence structure, more formal expressions.

DONE: The entire paragraph has been rewritten. The two main objectives of the study have been introduced.

Materials and methods: On line 207, Turkey analysis was written by mistake. It must be Tukey.

DONE: This typographical error has been corrected.

Results:

This section is very complex. It is not understood. It should be rewritten.

In particular, the titles describing figures 9,10,11 should be rewritten. Figures are incomprehensible.

DONE: titles of figures 9-11 has been correctly rewrited.

It should be stated in the figure titles that these images are SEM images.

DONE: It has been indicated that these images correspond to SEM micrographs.

The naming on the figures is confusing.

DONE: Naming of figures has been reviewed.

Discussion:

The titles of Figures 13-18 should be rewritten. Please write more clearly.

            DONE: titles of figures 13-18 has been correctly rewrited.

References: 

Reference 1, 6, 8, 10, 19, 35 and 40 are very old. Please replace what you can with new ones.

The references have been changed according to the reviewer.

Reviewer 2 Report

This paper is a very well-conducted study on Differences between the fittings of dental prostheses produced 2 

by CAD-CAM and Laser sintering processes. 

The paper structure and the overall content is good. 

Nevertheless, I suggest some improvements to be performed before this manuscript can be considered suitable for publication.

Lines 75-77 the authors wrote:

“In the field of restorative materials, ceramic materials such as yttria-stabilized zirconia, thermosetting polymers such as PEEK and metals that no longer contain nickel, which could cause allergic reactions, have appeared.”

Please support this sentence (every material) with proper references.

For example https://doi.org/10.3390/ma15217834

Lines 85-6 the authors wrote:

“This type of metal–ceramic restoration is also often referred to as a ceramo-metal, or porcelain infused-to-metal, restoration. “

Are there any papers that use this “naming”?

Materials and methods: The authors sometimes put the city of the product company. Sometimes not. Please standardize the document.

The authors have reported several (10,000) measurements. There is no description on how these measurements were performed. A description should be added.

The authors could add a sentence that could outline a limitation of this study. Interface analysis was performed statically. A simulation close to the clinical scenario can be obtained before and after cyclic fatigue. To overcome this limitation, the authors could therefore add a sentence like the following:

“One of the limitations of the current study is the lack of cyclic fatigue that could outline even more the adaptation of the investigated models”.

To support this sentence, the authors could cite the following reference: https://doi.org/10.1111/jerd.12837 

The authors should also mention that this experimental preparation design is a deep horizontal one. The authors should outline possible differences with vertical preparations.

Author Response

REVIEWER 2

Dear Reviewer,

Thanks for taking the time to review our manuscript and suggest to us to improve our work by providing a lot more detail. We have done so, and we are now submitting a manuscript that not only addresses the points you specifically raised but also many others that we have considered in order to deliver what we think is a much-improved version of our work. This version includes more paragraphs, figure, English grammar revisions in all main sections, new references. Thanks a lot. We are looking forward to your comments.

Sincerely,

Francisco-Javier Gil Mur

Comments and Suggestions for Authors

This paper is a very well-conducted study on Differences between the fittings of dental prostheses produced 2 by CAD-CAM and Laser sintering processes. 

The paper structure and the overall content is good. 

Nevertheless, I suggest some improvements to be performed before this manuscript can be considered suitable for publication.

Lines 75-77 the authors wrote:

“In the field of restorative materials, ceramic materials such as yttria-stabilized zirconia, thermosetting polymers such as PEEK and metals that no longer contain nickel, which could cause allergic reactions, have appeared.”

Please support this sentence (every material) with proper references.

For example https://doi.org/10.3390/ma15217834

The paper has been referenced in the text,

Lines 85-86 the authors wrote:

“This type of metal–ceramic restoration is also often referred to as a ceramo-metal, or porcelain infused-to-metal, restoration. “

Are there any papers that use this “naming”?

You are right. Ceramo-metal is unusual, and we have removed it from the text. Metal-ceramic and porcelain-fused-to-metal are common. In this last expression there was an error that has been corrected. I attach two references of two highly cited papers with these expressions in the titles.

Gregory Wall, Dale L. Cipra,

ALTERNATIVE CROWN SYSTEMS: Is the Metal-Ceramic Crown Always the Restoration of Choice?,

Dental Clinics of North America,

Volume 36, Issue 3,

1992,

Pages 765-782,

ISSN 0011-8532,

Herbert T. Shillingburg, Sumiya Hobo, Donald W. Fisher,

Preparation design and margin distortion in porcelain-fused-to-metal restorations,

The Journal of Prosthetic Dentistry,

Volume 29, Issue 3,

1973,

Pages 276-284,

Materials and methods:

The authors sometimes put the city of the product company. Sometimes not. Please standardize the document.

Done

The authors have reported several (10,000) measurements. There is no description on how these measurements were performed. A description should be added.

The authors have added a paragraph in materials and methods to clarify this point in agreement with the reviewer.

The authors could add a sentence that could outline a limitation of this study. Interface analysis was performed statically. A simulation close to the clinical scenario can be obtained before and after cyclic fatigue. To overcome this limitation, the authors could therefore add a sentence like the following:

“One of the limitations of the current study is the lack of cyclic fatigue that could outline even more the adaptation of the investigated models”.

To support this sentence, the authors could cite the following reference: https://doi.org/10.1111/jerd.12837 

The authors have added in the discussion the limitations of the study and have cited and referenced the interesting paper suggested by the reviewer.

The authors should also mention that this experimental preparation design is a deep horizontal one. The authors should outline possible differences with vertical preparations.

The authors have added this aspect in the abstract and materials and method. Thank you for your important suggestion.

Reviewer 3 Report

Abstract: - Please use a - between Cad-CAM through the manuscript or a /   - Please use . not , in number   - Please add the statistical analysis   - Please add a conclusion which reflects the findings   Introduction: -  Please clarify in details each used method   - CrCoMo, please add the complete name before abbreviation   - What is the originality of the study   - please add a null hypothesis   Methods: - Table 1: please add the complete name of each element   - Why only 10 samples for the control group?   - any sample size test?   - Please more details of each preparation method: Laser, Casting and Cad/Cam, machines, company....   - Please add a space between the number and its unit   - L165: mm2 please correct   - Please use scanning electron microscope and add more details about the preparation of samples to be analyzed with the microscope   - Which software was used to the statistical tests, and define the use of each test   Results: - Please use some arrows in SEM images to indicate the gaps   - Figure 12: what are X, + and the other indications? please more details in figure legend   Discussion: - There is no clear comparison between these findings and previous similar studies   - Please add the limitations and perspectives   - What about the EDX analysis, no info in the methods   Conclusions: - good but should reflect more the results   References: - Follow MDPI style 

Author Response

REVIEWER 3

Dear Reviewer,

Thanks for taking the time to review our manuscript and suggest to us to improve our work by providing a lot more detail. We have done so, and we are now submitting a manuscript that not only addresses the points you specifically raised but also many others that we have considered in order to deliver what we think is a much-improved version of our work. This version includes more paragraphs, figure, English grammar revisions in all main sections, new references. Thanks a lot. We are looking forward to your comments.

Sincerely,

Francisco-Javier Gil Mur

Comments and Suggestions for Authors

Abstract: 

- Please use a - between Cad-CAM through the manuscript or a /   

DONE: This typographical error has been corrected.

- Please use . not , in number  

DONE: This typographical error has been corrected. (Cad-CAM)

 - Please add the statistical analysis   

DONE: Statistical analysis has been introduced in abstract.

- Please add a conclusion which reflects the findings   

DONE: The conclusions section has been carefully revised and duly expanded following the indications provided by the reviewer. In addition, the conclusions section of the abstract has been completed.

Introduction: 

-  Please clarify in details each used method   

DONE

- CrCoMo, please add the complete name before abbreviation   

The authors have introduced the complete name of CrCo in the text

- What is the originality of the study 

A new paragraph has been introduced at the end of introduction

- please add a null hypothesis   

DONE: Both the hypothesis and the null hypothesis have been introduced in the text, following the reviewer's recommendations.

Methods: 

- Table 1: please add the complete name of each element 

DONE  

- Why only 10 samples for the control group?   

The casting method is currently rarely applied and in this paper we want to see the differences between CAD CAM and laser sintering. We have put 10 samples of known casting to give a reference of the traditional method but comparing with casting is not the main objective of this paper.

- any sample size test?   

DONE: The text has been revised and a more accurate explanation of the number of samples analyzed at each manufacturing site evaluated has been added.

“A total of 70 components were analyzed, at a rate of 10 components for each manufacturing plant evaluated. Ten cast components were tested as a control group, together with 30 laser sintering components and 30 CAD-CAM components. The components formed by CAD-CAM and LS were manufactured at a rate of 10 components for each of the 3 different manufacturing sites used for each technique”.

- Please more details of each preparation method: Laser, Casting and Cad/Cam, machines, company....   

DONE

- Please add a space between the number and its unit   

DONE: These typographical errors has been corrected.

- L165: mm2 please correct   

DONE: This typographical error has been corrected.

- Please use scanning electron microscope and add more details about the preparation of samples to be analyzed with the microscope   

DONE

- Which software was used to the statistical tests and define the use of each test.

DONE: The data concerning the statistical software used for the analysis of results as well as the statistical tests used for this purpose have been introduced in the materials and methods section of the article.

Results: 

- Please use some arrows in SEM images to indicate the gaps   

We cannot point them out because the distances measured in each sample are very large and we think it could confuse the reader.

- Figure 12: what are X, + and the other indications? please more details in figure legend   Discussion: 

The meaning has been introduced in the figure legend

- There is no clear comparison between these findings and previous similar studies   

- Please add the limitations and perspectives   

DONE. The last paragraph in Discussion

- What about the EDX analysis, no info in the methods  

DONE: An explanation of the EDX probe has been added to the materials and methods section of the article.

Conclusions: 

- good but should reflect more the results 

DONE: The conclusions section has been carefully revised and duly expanded following the indications provided by the reviewer.

References: - Follow MDPI style 

DONE

Reviewer 4 Report

The authors have performed a very interesting study on fittings of dental prostheses produced by CAD-CAM and Laser sintering processes. However, there are a few concerns and suggestions related to the work:

- Please mention the statistical methods used and the value of significance in the abstract

- The authors should write more about CAD/CAM and laser sintering in the introduction part. For CAD/CAM authors may use the following paper:

Khan AA, Fareed MA, Alshehri AH, Aldegheishem A, Alharthi R, Saadaldin SA, Zafar MS. Mechanical Properties of the Modified Denture Base Materials and Polymerization Methods: A Systematic Review. International Journal of Molecular Sciences. 2022 May 20;23(10):5737.

- What is the hypothesis of the research? Please mention

- In the last paragraph before RESULTS, the authors have mentioned about ANOVA tables. Where are they in the RESULTS?

- There are so many figures in the manuscript. It would be ideal if the authors merge some of the figures such as merging the three figures related to supra structure (Fig. 13, 14 & 16) to make it as one figure.

- Please write about the limitations of this research

Author Response

REVIEWER 4

Dear Reviewer,

Thanks for taking the time to review our manuscript and suggest to us to improve our work by providing a lot more detail. We have done so, and we are now submitting a manuscript that not only addresses the points you specifically raised but also many others that we have considered in order to deliver what we think is a much-improved version of our work. This version includes more paragraphs, figure, English grammar revisions in all main sections, new references. Thanks a lot. We are looking forward to your comments.

Sincerely,

Francisco-Javier Gil Mur

Comments and Suggestions for Authors

The authors have performed a very interesting study on fittings of dental prostheses produced by CAD-CAM and Laser sintering processes. However, there are a few concerns and suggestions related to the work:

- Please mention the statistical methods used and the value of significance in the abstract

DONE: Statistical method and value of significance has been introduced in the abstract.

- The authors should write more about CAD/CAM and laser sintering in the introduction part. For CAD/CAM authors may use the following paper:

Khan AA, Fareed MA, Alshehri AH, Aldegheishem A, Alharthi R, Saadaldin SA, Zafar MS. Mechanical Properties of the Modified Denture Base Materials and Polymerization Methods: A Systematic Review. International Journal of Molecular Sciences. 2022 May 20;23(10):5737.

- What is the hypothesis of the research? Please mention

DONE: Both the hypothesis and the null hypothesis have been introduced in the text, following the reviewer's recommendations.

- In the last paragraph before RESULTS, the authors have mentioned about ANOVA tables. Where are they in the RESULTS?

The results of the statistical studies are in the Figure 12 where the statistical differences significances have been marked in the columns with a p-value<0.05

Round 2

Reviewer 1 Report

The corrections given were made by the authors. Thanks to the authors. As such, it is acceptable.

Author Response

Thank you very much for your help. We are revised the English again. Thanks a lot

Reviewer 3 Report

Dear authors,

thanks for the revision, the paper is now ready for publication from my side.

Best wishes and have a nice weekend!

Author Response

Thank you very much for your help. 

Reviewer 4 Report

Thank you for addressing the concerns and comments. I have no more

Author Response

Thank you very much for your help-